# Evaluation of the Impact of Near-Infrared Multiwavelength Locked System Laser Therapy on Skin Microbiome in Atopic Dogs

**DOI:** 10.3390/ani14060906

**Published:** 2024-03-14

**Authors:** Sara Muñoz Declara, Aldo D’Alessandro, Agnese Gori, Benedetta Cerasuolo, Sonia Renzi, Michele Berlanda, Eric Zini, Monica Monici, Duccio Cavalieri, Giordana Zanna

**Affiliations:** 1Anicura Istituto Veterinario di Novara, Strada Provinciale 9, 28060 Granozzo con Monticello, NO, Italy; sara.munoz@anicura.it (S.M.D.); eric.zini@anicura.it (E.Z.); 2Department of Biology, University of Florence, Via Madonna del Piano 6, 50019 Sesto Fiorentino, FI, Italy; aldo.dalessandro@unifi.it (A.D.); benedetta.cerasuolo@unifi.it (B.C.); sonia.renzi@unifi.it (S.R.);; 3Department of Animal Medicine, Production and Health, University of Padova, Viale dell’Università 16, 35020 Legnaro, PD, Italy; michele.berlanda@unipd.it; 4Clinic for Small Animal Internal Medicine, Vetsuisse Faculty, University of Zurich, Winterthurerstrasse 260, 8057 Zurich, Switzerland

**Keywords:** photobiomodulation, microbiome, canine atopic dermatitis, mycobiota, MLS^®^ laser therapy

## Abstract

**Simple Summary:**

Photobiomodulation is the use of red or near-infrared light to produce beneficial effects in living biological tissue. Several studies in humans have demonstrated the beneficial effects of photobiomodulation in improving tissue healing, reducing inflammation and edema, restoring blood flow, and inducing analgesia. Growing evidence suggests that photobiomodulation may influence the skin microbiome. The dog skin microbiome has been previously investigated in healthy and allergic dogs, but only limited information is available on this topic. The aim of this research was to evaluate the effect of photobiomodulation on the skin microbiome of atopic dogs by employing a high-power, dual-wavelength near-infrared laser source (Multiwavelength Locked System laser, MLS^®^). The results showed that while the microbiome composition and diversity are not significantly affected, photobiomodulation could induce an overall reduction in specific bacterial species, in particular *Staphylococcus*, that represent a major pathogenic skin strain. This observation could be a potential jumpstart for further analysis in larger populations to test the beneficial effects of photobiomodulation in dog skin.

**Abstract:**

Photobiomodulation (PBM) is a newly adopted consensus term to replace the therapeutic application of low-level laser therapy. It has been suggested that PMB influences the microbiome which, in turn, has increasingly been shown to be linked with health and disease. Even though the use of PBM has also grown dramatically in veterinary medicine, there is still a lack of evidence supporting its effect in vivo. Our objective was to investigate the impact of a dual-wavelength near-infrared laser source (Multiwavelength Locked Laser System, MLS^®^) on the skin microbiome in atopic dogs. Twenty adult-client-owned atopic dogs were enrolled in the study. The dogs were treated with MLS^®^ laser therapy on one half of the abdominal region, whereas the contralateral side was left untreated and served as a control. Skin microbiome samples were collected before and after MLS^®^ treatments, and then subjected to NGS-based ITS and 16S rRNA analysis. The results showed that while microbiome composition and diversity were not significantly affected, PBM could play a role in modulating the abundance of specific bacterial species, in particular *Staphylococcus*, that represent a major skin pathogenic strain. To the best of the authors’ knowledge, this is the first study to investigate the potential impact of MLS^®^ laser therapy on the skin microbiome in atopic dogs.

## 1. Introduction

The principle of the laser, an acronym for light amplification by stimulated emission of radiation, dates back to 1916, when Albert Einstein first described the theory of stimulated emission. It was not until 1960, however, that the first working laser was created by Theodore Maiman at Hughes Aircraft [1,2,3].

In 1967, Dr. Endre Mester and his colleagues used a murine model to investigate the oncological safety of low-level laser irradiation at the red/near-infrared end of the spectrum. They found no evidence of skin dysplasia but incidentally observed enhanced hair regrowth [4,5]. This observed effect was termed “laser biostimulation”. They later demonstrated accelerated wound healing following exposure to low-level near-infrared laser light. Based on these results, the study of the biological effects of low-power lasers began, and in 1974, the biostimulation effect was first applied in clinical practice [4,6].

Over the last 30 years, lasers have become a well-established method of light energy delivery to the skin for different therapeutic purposes [4]. Laser irradiation can induce a photobiomodulatory effect on cells and tissues, which can help, for example, to mitigate inflammation and pain in several clinical scenarios [7,8,9]. Although the umbrella term photobiomodulation (PBM) has been assigned many definitions, PBM is best described as the intentional use of red or near-infrared (NIR) light to produce beneficial effects in living biological tissue [4,10,11]. The first postulate of photobiology states that to have any biological effect, photons of light must be absorbed by chromophores located within the tissue. According to recent experimental evidence, the most important chromophore is the cytochrome c oxidase (CCO) unit IV of the mitochondrial respiratory chain [4,12,13]. The absorption of energy by CCO is the primary interaction triggering the PBM effect and it results in enhanced cell signaling, mitochondrial ATP production, and growth factor synthesis, as well as the attenuation of oxidative stress [4,6,14]. All these events are normally influenced by physical irradiation parameters, such as energy density, wavelength, power density, emission mode (pulsed or continuous wave), irradiation time, as well as by clinical irradiation parameters, such as irradiated area or point numbers, application technique, and treatment frequency [15].

The beneficial effects of PBM in reducing inflammation and edema, restoring blood flow, improving tissue healing, and inducing analgesia have been widely demonstrated in humans [16,17,18]. Moreover, there is also growing evidence that PBM may influence the microbiome [19,20,21,22,23,24]. Interestingly, Bicknell and colleagues demonstrated that PBM delivered to the abdomen of mice produced a significant change in the gut microbiome and, in humans, they observed a change in the microbiome in Parkinson patients after PBM treatment [21,22].

Photobiomodulation has also been extensively studied in veterinary medicine, with wound healing as the main field of its application [1,2]. The clinical efficacy of PBM has been evaluated on localized lesions in canine atopic dermatitis (CAD), with, however, poor results [25]. CAD is a hereditary, pruritic, and predominantly T-cell-driven inflammatory skin syndrome, involving a complex interplay between skin barrier abnormalities, allergen sensitization, and microbial dysbiosis [26,27,28]. Differences in microbial richness between healthy and allergic dogs have also been recently described [29,30]. However, whether the dysbiosis is caused by or is a consequence of the CAD status is still undetermined, and at present, it is not possible to elucidate the pathogenetic role of dysbiosis in CAD [26,27,28].

Against this background, the objective of this research was to evaluate the effect of PBM on the composition and abundance of the skin microbiome of atopic dogs by employing a high-power, dual-wavelength NIR laser source (Multiwavelength Locked System laser, MLS^®^). The aim was to pave the way for innovative treatments in inflammatory skin diseases such as CAD.

## 2. Materials and Methods

Ethics Statement

This study was designed as a prospective, intra-individual study, with each dog as its own control. It was approved by the ethics committee in charge of animal welfare of the University of Padova (Italy) with the protocol number OPBA 29/2020, with the sampling period extending from October 2021 to November 2022. Dog owners signed a written informed consent form before inclusion and could withdraw it at any time.

### 2.1. Study Population

Twenty adult-client-owned atopic dogs were enrolled in the study. Eight of the twenty dogs were completely outdoor, and the others both indoor and outdoor. A proportion of 12/20 were spayed females, 6/20 were intact males, and 2/6 neutered males. A proportion of 13/20 of the dogs were mixed breeds. The remaining subjects were 1 Beagle, 1 Poodle, 1 Staffordshire terrier, 1 West Highland White Terrier, 1 Labrador Retriever, 1 Cocker Spaniel, and 1 French Bulldog. Twelve dogs co-inhabited with other animals (dogs and/or cats). The age range of the patients was from 2 to 11 years old with a median age of 6 years old.

Dogs were selected according to Favrot’s diagnostic criteria [31], with other pruritic dermatoses such as flea allergy dermatitis, cutaneous adverse food reactions, and/or sarcoptic mange excluded. Indeed, prior to sampling, all dogs in the study (also including dogs and cats of the household when present) had received monthly treatment with an oral isoxazoline and had been strictly fed a hydrolyzed protein diet for a minimum of 8 weeks and then included only when no improvement in the clinical signs was observed during the trial. Moreover, dogs could not (i) display systemic signs of concomitant diseases; (ii) if females, be pregnant or lactating; (iii) have received topical or systemic anti-inflammatory or immunomodulatory drugs including glucocorticoids, oclacitinib, or cyclosporine, and/or topical or systemic antimicrobials for at least 30 days; or (iv) have been bathed with antibacterial or antifungal treatments for at least 7 days prior to sample collection. Dogs undergoing allergen-specific immunotherapy and/or canine monoclonal antibody treatment were also excluded.

Dogs were finally included when no other gross lesions on dermatological examination besides erythema of feet, ear pinnae, and ventral region associated with pruritus were detected.

### 2.2. Study Protocol (Sample Collection and MLS^®^ Laser Treatment)

The study design protocol was established for each dog for 6 days and was as follows. Day 1: Screening visit and collection of skin swabs from both sides of the abdomen for microbiome analysis. One side of the abdomen was treated with MLS^®^ laser therapy, whereas the contralateral side of the abdomen was not treated and served as control. The same protocol was used on Day 3, whereas on Day 5, a third MLS^®^ treatment with no sample collection was performed, and on Day 6, a third collection of skin swabs without MLS^®^ treatment was performed.

MLS^®^ laser (M-VET^®^, Asa S.r.l, Vicenza, Italy) is a Class IV laser with a high peak power, which is characterized by a synchronized emission of pulsed and continuous waveforms. The two modules have different wavelengths, peak power, and emission modes. The first one is a pulsed diode laser, emitting at 905 nm, with peak optical power 360 W; each pulse is composed of a pulse train (single pulse width = 100 ns, maximum frequency 40 kHz), thus varying the average power delivered to the tissue. The frequency of the pulse trains is in the range 10–40 kHz. The second laser diode (808 nm) operates in continuous mode (P 4 W) or in pulsed mode (pulses repetition rate 1–2000 Hz), with mean optical power output of 3.3 W and 1.65 W, respectively [32].

In the present study, the MLS^®^ treatment was applied in scanning mode on half of the abdomen of the dogs. The dual wavelengths were emitted in a single handpiece with 2 cm diameter. The average treatment area was 50 cm^2^. The laser treatment was established in the “Atopic dermatitis” program of the manufacturer. The parameters were as follows: 4.56 J/cm^2^ energy density, intensity 70%, 18 Hz frequency, time for treatment 3 min and 14 s, and 228.15 J total delivered energy.

Some precautions and procedures were performed such as a laser probe being applied perpendicularly over the patient’s skin to avoid reflection, and the utilization of laser goggles. The laser was calibrated by the manufacturing company before starting the study.

Both areas of the abdomen were swabbed for 15 s in a 50 cm^2^ area using a sterile collection method. Before sampling, the swabs were soaked in RNAlater solution and after having swiped them on the dog’s skin, they were inserted into “PowerBead Pro Tubes” (provided by the chosen extraction kit DNeasy^®^ PowerSoil^®^ Pro Kit), filled with 250 µL of RNAlater solution. The samples were then sent to the Microbiology Laboratory of the Biology Department of the University of Florence (Italy) to perform the molecular diagnostics.

### 2.3. DNA Extraction and Sequencing

Out of 20 patients enrolled in the study, one dog from the bacterial analysis and 11 dogs from fungal analysis were removed from the study because of a low amount of DNA in the samples.

The swabs contained in the “PowerBead Pro Tubes” were processed using the protocol indicated by the DNeasy PowerSoil Kit (Qiagen, Hilden, Germany) in order to extract the DNA that was present in the samples, and the quality of the extraction was assessed through 1% agarose gel electrophoresis. The Qubit 4 Fluorometer (Thermo Fisher Scientific, Waltham, MA, USA) and the 1x dsDNA High Sensitivity kit were used to measure DNA concentration prior to subsequent analysis.

Amplification of both the target regions, ITS1 (ITS1f: 5′-CTTGGTCATTTAGAGGAAGTAA-3′ and ITS2r: 5′-GCTGCGTTCTTCATCGATGC-3′) and V3–V4 hypervariable regions of the 16 rRNA gene (341f: 5′-CCTACGGGNGGCWGCAG-3′ and 805r: 5′-GACTACNVGGGTWTCTAATCC-3′), respectively, for fungal and bacterial communities, was performed on each sample using specific primer sets provided with overhang Illumina adapters [33,34]. Amplicon library preparation was carried out in accordance with Illumina protocols [35]. Paired-end 2 × 300 bp sequencing was performed on an Illumina MiSeq instrument (Illumina Inc., San Diego, CA, USA) with the MiSeq Reagent Kit v3 (600 cycle) [36,37,38].

### 2.4. Post-Run Analyses

After sequencing, demultiplexed sequences were quality-filtered and cleaned using Quantitative Insights Into Microbial Ecology (QIIME2) software (v. 2023.2) [39]. The CUTADAPT plugin was used to remove primers, and the Divisive Amplicon Denoising Algorithm (DADA2) plugin [40] was used for read merging, to remove chimeras, and to determine Amplicon sequence variants (ASVs). Merging of reads was performed with a quality score threshold of 20 and a max error of 2. Patients with at least one sample which did not pass the filtering steps were removed from the analysis. Taxonomic annotation was performed using RESCRIPt plugin (v.2023.2.0) [41] with Silva 138.1 database [42] for 16S sequences, and UNITE database [43] for ITS sequences. Further analyses were handled and graphically represented in R software (v.4.2.2) [44], using the packages phyloseq v.1.42.0 [45], microbiome v.1.21.1 [46], vegan v.2.6.4 [47], tidyverse v.2.0.0 [48], and ggplot2 v.3.4.2 [49]. ggpubr v.0.6.0 [50] and patchwork v.1.1.2 [51] were used for data visualization.

Both bacterial and fungal datasets were manually further cleaned to include only bacterial and fungal taxa, respectively.

### 2.5. Diversity Indices

Alpha diversity was evaluated using inverse Simpson index, while pairwise Wilcoxon test was used to detect significant differences between samples’ diversities. A distance matrix of ASVs based on Bray–Curtis distances was used to explore microbial communities’ differences among samples through a Non-Metric Multidimensional Scaling (NMDS) approach. Significance of differences in microbial composition among different factors was evaluated through Permutational Analysis of Variance (PERMANOVA).

### 2.6. Differential Abundance Testing

Differentially abundant taxa were detected using the Analysis of Compositions of Microbiomes with Bias Correction (ANCOMBC) package (v.2.0.2) [52] at two taxonomic levels: genus and family. This analysis was performed with a q-value threshold of 0.05, and structural zeros were removed.

## 3. Results

### 3.1. Read Number and Quality

Of the initial 9,819,485 reads, 5,302,353 were kept (~54% of the total, a mean number of ~55.000 reads per sample with a maximum number of 142,982 and a minimum number of 20,600 reads) after the filtering step.

### 3.2. Community Composition

From the observations of the relative abundance of the taxa, the most abundant bacterial taxon across all the samples was Staphylococcaceae, almost entirely represented by the genus *Staphylococcus*. Amplicon sequence variants belonging to this family represented ~24.19% of the total, with a maximum abundance of ~80% in two of the dogs (i.e., patients 9 and 17). Following this, Lactobacillaceae accounted for 5.57% of the total community, Micrococcaceae for 5.38%, Moraxellaceae for 4.81%, and Bifidobacetriaceae for 4.72%. All of these taxa showed an increase in individuals with a lower abundance of *Streptococcus*.

Fungal communities were shown to be highly dominated by a single family. Pleosporaceae represented 66.49% of the total community, with a lower abundance in indoor individuals. The following families in terms of abundance were Didymosphaeriaceae, representing 8.31%, Saccharomycetaceae 4.79%, Saccotheciaceae 2.32%, and Cladosporiaceae 1.49% of the total community. All of these families showed variable abundances but were not connected to the presence of Pleosporaceae. However, Didymosphaeriaceae showed a higher abundance in the indoor individuals.

The relative abundance of the most abundant families and genera is shown in Figure 1.

### 3.3. Laser Treatment Reduces the Presence of Infection-Linked Taxa

The differential abundance analysis performed using an ANCOMBC with a significance threshold of 0.05 did not reveal any statistically significant change in microbial abundances linked with laser treatment. However, a trend was observed when comparing the relative abundance of the bacterial and fungal families over time in the treated and non-treated sides. Indeed, genera like *Staphylococcus*, which is linked to skin infections in dogs, decreased more in the treated side of the skin during the course of the experiment, while presenting a slower decrease or even an increase in the non-treated side (Figure 2).

Other taxa (e.g., Corynebacterium) did not show a trend based on the treatment, but rather correlated to the individual’s condition.

### 3.4. Breed Influences the Abundance of a Limited Number of Taxa

Taking into account other factors (e.g., breed, environment, time), the differential abundance analysis revealed a very low number of phyla, families, and genera which significantly changed across the dataset, and these taxa showed a different abundance pattern based only on the dog’s breed.

At the phylum level, Actinobacteria, Chloroflexi, and Proteobacteria showed a greater mean abundance in the mixed breed dogs. The same behavior was observed at the family level, with Aerococcaceae, Corynebacteriaceae, Microbacteriaceae, Micrococcaceae, and Rhodobacteraceae having a higher mean abundance in the mixed breeds, and at the genus level, with *Kocuria* also showing a significantly greater abundance in the mixed breeds (Figure 3a).

For the fungal dataset, significant differences between breeds were only found at the family and genus levels. For both taxonomic levels, only one taxon was found to be significant. The genus *Pseudopithomyces* and family Didymosphaeriaceae were both observed to increase in the Cocker Spaniels, Labrador Retrievers and, especially, the mixed breeds. Their abundance was close to zero in all the other breeds (Figure 3b).

### 3.5. Breed and Background Influence Microbial Diversity

The differences in the microbial community diversity were evaluated using the inverse Simpson index. The fungal diversity was discovered to be ~10 times lower than the bacterial diversity, and no significant differences were found between the indoor and outdoor samples, between samples at different time points (Figure 4b,d), and between the treated and non-treated dogs (Appendix A). However, microbial diversities were found to change according to the breed of the dogs (Figure 4a,c).

To calculate the statistical significance between the Simpson values across the different breeds, the Wilcoxon test was used. For bacteria, the values obtained for the West Highland White terrier significantly differed from those obtained for all the other breeds (*p* < 0.01), while the values obtained for the Labrador Retrievers were significantly different from the mixed breeds, French Bulldogs, and Beagles (*p* = 0.007, *p* = 0.007, and *p* = 0.012 respectively) (Figure 5a). For the fungal dataset, values from the French Bulldogs were significantly different from the Labrador Retrievers, mixed breeds, and Cocker Spaniels (*p* = 0.024, *p* = 0.001, and *p* = 0.02 respectively), while the Cocker Spaniels were significantly different from the Labrador Retrievers and mixed breeds (*p* = 0.024 and *p* = 0.013 respectively) (Figure 5b).

The boxplots of the Shannon and Chao1 diversities, and the relative heatmap of the results obtained from the Wilcoxon test, are shown in Appendix A.

The diversity values obtained with different grouping factors (e.g., time after treatment, background, sex) did not produce any significant comparison.

Beta diversity was explored using Bray–Curtis distances with an NMDS approach. As for the alpha diversity, the beta diversity analysis revealed that the only two factors influencing the community composition were the dog’s breed and, especially, their background, with time and treatment not showing any clustering (Appendix A). Indeed, the breed was shown to have a role in modulating both bacterial and fungal communities, since a partial clustering of the samples was visible (Figure 6).

However, the most visible clustering of the samples was based on the dogs’ background (Figure 7).

### 3.6. Beta Diversity Is Influenced by Both Breed and Background

The correlation of the community composition with the breed and breeding site was confirmed by the PERMANOVA analysis. For the bacterial communities, the background accounted for 13.89% of the total variance (R2 = 0.1389) and produced an F-value of 20.39 and a *p*-value of 9.999 × 10^−5^, while the breed produced an R2 of 0.2277, an F-value of 4.77, and a *p*-value of 9.999 × 10^−5^.

For the fungal communities, the background accounted for 20.14% of the total variance (R2 = 0.2014) and produced an F-value of 17.12 and a *p*-value of 9.999 × 10^−5^ (***), while the breed produced an R2 of 8.9507, an F-value of 8.95, and a *p*-value of 9.999 × 10^−5^ (***).

## 4. Discussion

In the present study, we examined the effects of MLS^®^ laser therapy (M-VET^®^, Asa S.r.l, Vicenza, Italy), on the composition and abundance of the skin microbiome in atopic dogs. The microbiome composition and diversity were not significantly affected, whereas it was demonstrated that PMB could play a role in modulating the abundance of specific bacterial species such as *Staphylococcus pseudintermedius*, the most frequent bacterial pathogen in clinical canine specimens [53].

Over the past decade, research into the cutaneous microbiome of both healthy and atopic dogs has seen significant growth. Recent investigations have consistently shown that healthy dogs have a highly diverse skin microbiome, contrasting with the compromised diversity in those with atopic dermatitis [29,54,55]. Several studies reveal a common occurrence of cutaneous dysbiosis in dogs with atopic dermatitis, and this imbalance in the skin microbial population is associated not only with reduced diversity but also with a decline in beneficial bacteria [26,30,54,56,57,58]. Numerous factors such as interaction with the environment, anatomical location, and, possibly, the behavioral component could influence the composition of the microbiome [59,60,61].

In this prospective, intra-individual study, we first investigated different factors, including the individual’s breed and environment, to assess their potential influence on the microbiome composition. Our findings indicated that the distinction between dogs kept predominantly outdoors versus a combination of both indoors and outdoors did noticeably impact the bacterial or fungal diversity in our samples, similar to what has previously been described [54,57,59,60]. However, additional factors could modify the microbiome composition such as skin barrier alterations, immune response, genetic background, cohabitation with other animals and/or humans, and diet, among others [57,59,60,62,63,64,65,66,67].

Interestingly, in our results, the breed per se was identified as a significant factor affecting the microbiome composition. This result was supported by previous studies which have suggested a potential influence of the breed genetic variability on the skin microbial community [30,58]. Our findings invite future research on a larger number of animals per breed, since the skin microbiome has already been described as highly variable depending on the individual, and a larger number of individuals should be analyzed in order to generalize our conclusions [26,56,59,60].

In our study, the most abundant bacterial phyla were Firmicutes, Actinobacteria, and Proteobacteria, and the most abundant fungal phylum was Ascomycota. These results correlated with those of previous studies [29,57,68].

Several studies have demonstrated that atopic dogs are characterized by a general reduction in the microbiome diversity and an increase in the relative abundance of *S. pseudintermedius*. This shift has also been strongly correlated with the severity of clinical signs in CAD [69].

In the past, the potential effects of PBM on specific bacterial species both in vitro and in vivo has been demonstrated. For example, in an in vitro study, de Sousa et al. reported that the growth of bacteria such as *Pseudomonas aeruginosa*, *Escherichia coli*, and *Staphylococcus aureus*, which are commonly associated with infected ulcers, was inhibited using PBM.

Moreover, the efficacy of PBM in reducing *Staphylococcus aureus* colonization and promoting the healing of infected skin wounds has been demonstrated also in diabetic rats [70,71,72,73].

More recently, a placebo-controlled clinical trial was published in the field of veterinary medicine to investigate the effect of PBM on the bacterial load in traumatic wounds in dogs. The authors observed a statistically significant reduction in the average count of colony-forming units in the PBM-treated groups compared to the placebo group [74].

In humans, the effect of PBM using a long-pulsed alexandrite laser therapy on patients affected by rosacea to assess changes in skin microbiota composition has been recently published [75]. The authors demonstrated a positive treatment effectiveness with some impacts on the skin microbiota composition. In the field of veterinary medicine, there is only one study that has investigated the effect of phototherapy on the skin microbiome and skin barrier function in canine atopic dogs [76]. In this research, phototherapy was demonstrated to increase microbial diversity and decrease the relative abundance of *Staphylococcus pseudintermedius*.

When the effect of MLS^®^ laser therapy on the skin microbiome was compared intra-individually, no statistically significant results were obtained when looking at the diversity indexes. Nevertheless, we observed an increased diversity in taxa and a reduction in the relative abundance of *S. pseudintermedius* in the treated area of some patients.

In accordance with the above-mentioned studies, our results suggest a possible positive effect of MLS^®^ laser therapy on the reduction in *S. pseudintermedius*. This observation could be due to two possible mechanisms: (i) a reduction in the relative abundance of *S. pseudintermedius*, or (ii) a normalization of the entire microbiome, as proven in recent studies involving the use of antimicrobial therapy in CAD patients [57,69]. However, all these observations require further investigation and confirmation in a larger number of cases.

*Malassezia* spp. dysbiosis occurs in allergic dogs and has recently been demonstrated by Meason-Smith et al. [77]. Although *Malassezia* is one of the most dominant fungi genera reported on canine skin by culture [30], in the present study, it was detected with a very low relative abundance, with the exception of dog 11. This finding is consistent with what was recently reported by Chermprapai et al. [57]. However, the small sample size analyzed in our study did not allow us to draw a definitive conclusion, and further studies should be performed to investigate this issue.

Recent research has explored the potential of PBM to alter the microbiome as a therapeutic tool in various human diseases [17,19,22,78]. A recent study has highlighted PBM’s role in inhibiting the primary inflammatory response triggered by keratinocytes [79]. This discovery, coupled with other studies focusing on the gut–skin microbiome axis in human atopic dermatitis, that is now also being explored in veterinary medicine [58,80,81,82], underscores the importance of further investigating PBM’s impact on the skin microbiome in inflammatory skin conditions. The potential use of PBM as a non-invasive and effective tool warrants additional research in this context.

## 5. Conclusions

To the best of our knowledge, this is the first study to investigate the effects of MLS^®^ laser therapy (M-VET^®^, Asa S.r.l, Vicenza, Italy), with the following parameters: 4.56 J/cm^2^, intensity 70%, 18 Hz, time 3 min and 14 s, and 228.15 J total delivered energy, on the composition and abundance of the skin microbiome in atopic dogs. Although no effects on the microbiome composition and diversity were demonstrated, a decrease in the relative abundance of specific bacterial species, in particular *Staphylococcus*, was observed in the treated side of some of the atopic dogs. The insights from this cohort study suggest that PBM could represent a potential jumpstart for further analysis in larger populations to test the beneficial effects of PBM on dog skin.

## Figures and Tables

**Figure 1 animals-14-00906-f001:**
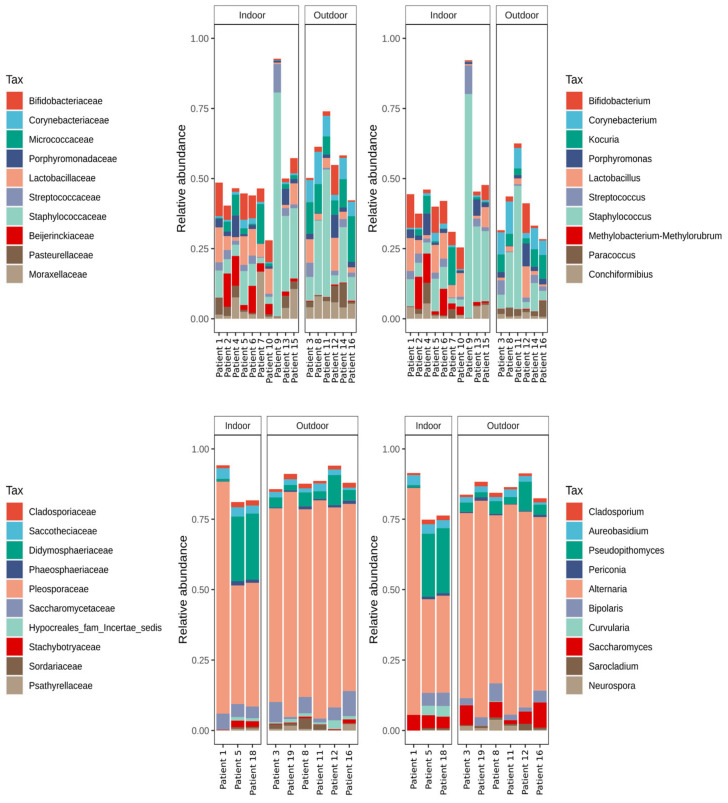
Bar plots of the 10 most abundant taxa. Plots a and b show the 10 most abundant families and genera across bacterial communities, plots c and d show the 10 most abundant families and genera in the fungal datasets.

**Figure 2 animals-14-00906-f002:**
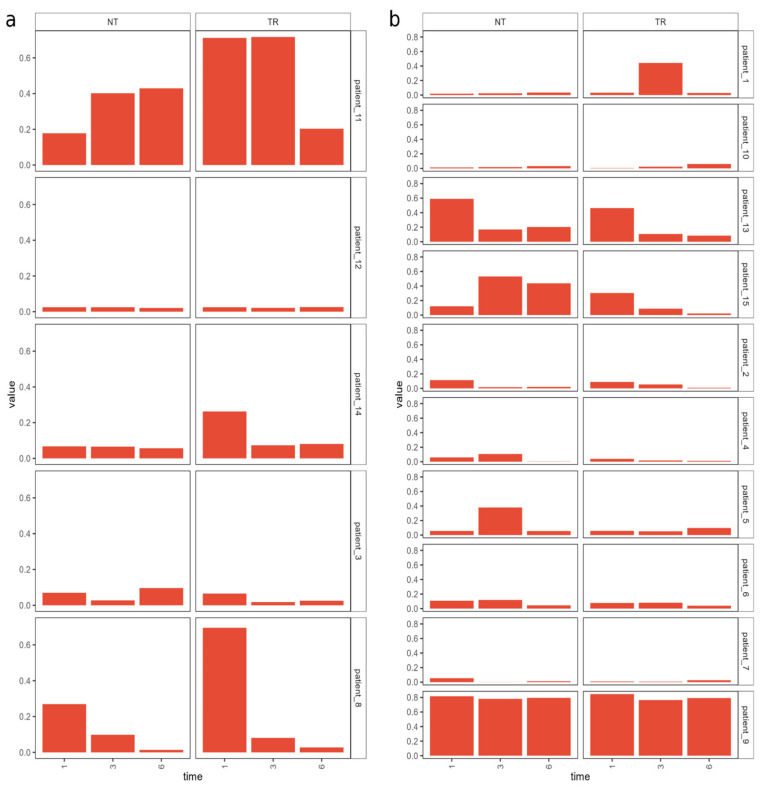
Bar plot of the relative abundance of genus *Staphylococcus* in outdoor (**a**) and indoor (**b**) patients.

**Figure 3 animals-14-00906-f003:**
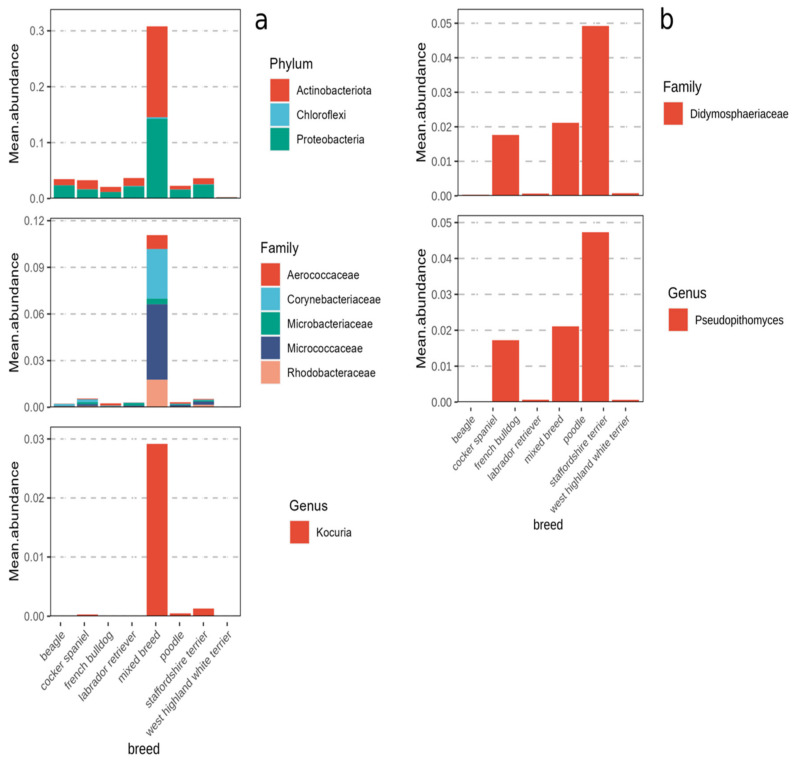
Bar plots of differentially abundant taxa in bacterial (**a**) and fungal (**b**) communities across breeds.

**Figure 4 animals-14-00906-f004:**
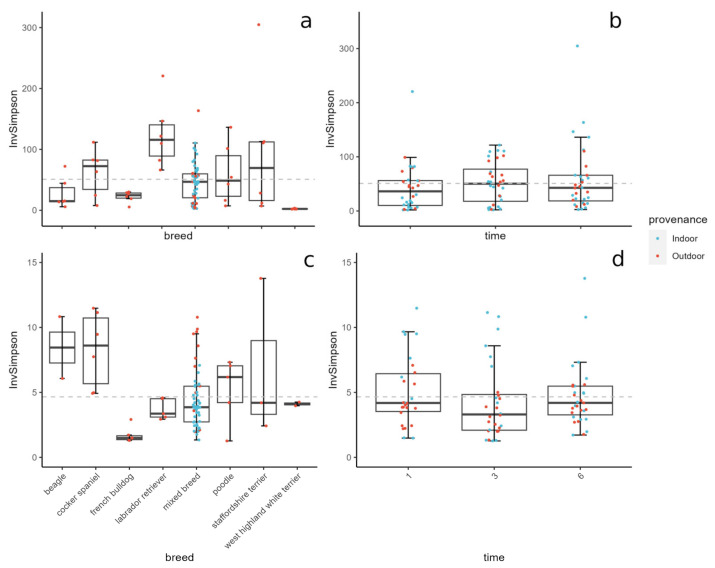
Boxplot of inverse Simpson’s diversity across breeds (**a**,**c**) and time after first treatment (**b**,**d**). On top are the bacterial data, on bottom are the fungal data. Colors indicate the dogs’ provenance.

**Figure 5 animals-14-00906-f005:**
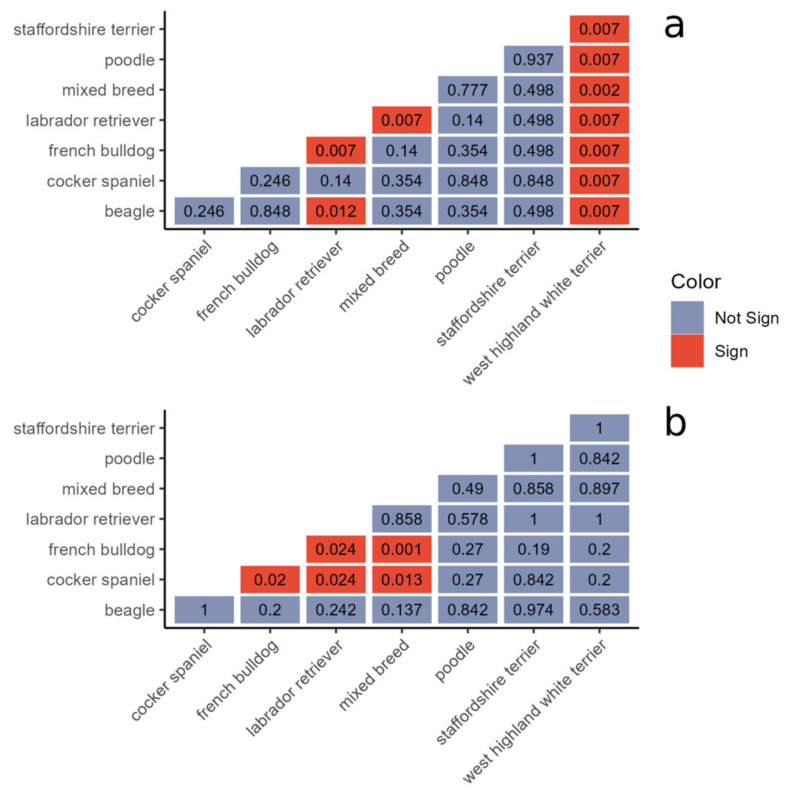
Results of Wilcoxon test between inverse Simpson values across bacterial (**a**) and fungal (**b**) datasets.

**Figure 6 animals-14-00906-f006:**
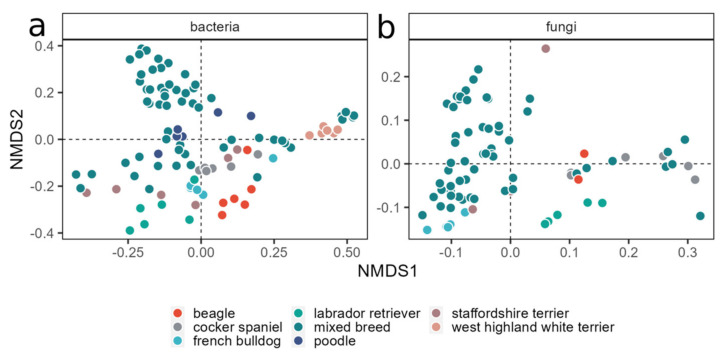
NMD plots of bacterial (**a**) and fungal (**b**) communities. For the bacterial dataset, the best solution was found with a stress value of 0.1555; for the fungal dataset, the stress value was 0.089. Colors indicate the dogs’ breed.

**Figure 7 animals-14-00906-f007:**
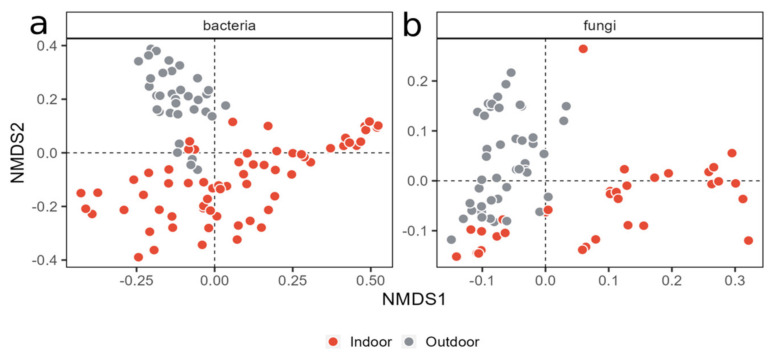
NMD plots of bacterial (**a**) and fungal (**b**) communities. Stress values were 0.1555 for the bacterial and 0.089 for the fungal dataset. Colors show the dogs’ provenance.

## Data Availability

The original contributions presented in the study are included in the article/Appendix A; further inquiries can be directed to the corresponding author/s.

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
