# Peer review of "Evaluation of the Impact of Near-Infrared Multiwavelength Locked System Laser Therapy on Skin Microbiome in Atopic Dogs"

_animals, 2024, doi:10.3390/ani14060906_

Round 1

Reviewer 1 Report

Comments and Suggestions for Authors

The authors claimed to investigate the impact of the MLS treatment on the skin microbiome in atopic dogs. The study brings new insights on the effect of this treatment and the novelty of the research is extremely interesting.

Although these premises, the manuscript has some minor edits that need to be addressed throughout the text, but most importantly, the results do not describe the aim of the research.

Line 23: change “induces” to “induced”.

Line 32: on THE skin microbiome.

Line 36: molecular analysis is too general.

Keywords: either microbiome or microbiota, also it is “mycobiota” and not micobiota.

Line 48: this sentence is confusing, revise the English form.

Line 54: take off the comma after (CCO).

Line 90: spayed.

Line 91: neutered.

Line 97-100: revise the entire sentence since it’s confusing.

Line 123: for each dog.

Line 147-148: missing pieces of information.

Line 150-154: revise English form.

Line 156: V3 and V3 regions of what? Missing information.

Line 161: Paired-end 2 x 300 bp is impossible for ITS1.

Line 165: Qiime2 is an abbreviation. Write the acronym.

Line 166: Dada2 is an abbreviation, and it’s missing the citation.

Line 185: ANCOMBC is an abbreviation?

MATERIAL AND METHODS:

There is a wrong definition of “placebo”. This word cannot be used to describe a control side.

RESULTS

The findings of this study provide a comprehensive analysis of the microbiome and mycobiome, detailing their composition and elucidating the influence of various factors such as breed and environment. However, there is a critical oversight regarding the specificity of the analyses conducted on samples collected from both treated and untreated sides of the abdomen, as well as across the three designated time points. Without clarification on this matter, the assumptions made by the authors may be deemed inaccurate or inadequately supported. It is imperative to emphasize that a thorough understanding of these factors can only be derived from analyses conducted at the initial time point, prior to any treatment interventions. Furthermore, the discussion on the effects of treatment appears disproportionately brief, considering it should be the primary focus of the study, as indicated by the authors themselves. Therefore, a more extensive examination of the treatment effects is warranted to fulfill the study's objectives effectively.

Regrettably, due to the aforementioned discrepancies and inadequacies in the analysis and focus of the study, I am unable to proceed with the revision of the article at this time. It is essential that these issues are addressed and clarified to ensure the scientific integrity and validity of the research findings. Once these concerns have been adequately resolved, I would be more than willing to revisit the revision process. 

Comments on the Quality of English Language

The manuscript is well written, only some minor edits need to be addressed. 

Reviewer 2 Report

Comments and Suggestions for Authors

Dear authors, thank you for your exciting article. I understand the hard work you have done and I hope my recommendations will be helpful. 

Lines 76-80: I am wondering if the first step would be to do this study in healthy (non-allergic dogs) and then in allergic dogs? Has anyone done this? Perhaps if there were non allergic animals to serve as control (for the disease) it would be beneficial. Or did the allergic dogs have pyoderma and your objective is to evaluate the microbiome of allergic dogs with pyoderma when treated with PBM? 

Line 83: was the study blinded? 

Line 148: University of XX. Since it is not a blinded review I think you can add the university's name. 

Line 178: Why did you evaluate only the inverse Simpson index? Most of the publications evaluate at least the Shannon index (+/- Chao-1 index).  I highly recommend reporting the Shannon index, except if there is a specific reason why you chose only this index. 

Results: I do not feel comfortable with the way the results are reported. The main objective was to evaluate the influence of the  PBM in the treated area vs the non-treated area. You should start with these results first which are your objective. Then you could report other results you found.

Figures 3 and 4. Please report the stress values alongside NMDS plots to enhance the interpretability of the results and allow readers to assess the reliability of the spatial configuration in the plot. 

Lines 228-235: Was Bonferoni-correction (or something similar) used for the  p values, to control for the increased risk of Type I errors due to multiple testing? Please clarify, and if not, please perform.

Lines 242 - 245: Did you perform cytology (that was something I wanted to ask for the M&M). Could some of the dogs with high abundance of Staphylococcus (Patient 9 and 17 for example) have a bacterial overgrowth or pyoderma?

Paragraph 3.5: As I mentioned above this should be the  first paragraph in the results section. Also, even if not significantly different, please provide images and the numbers for the alpha and beta diversity analysis. It is your main objective and the readers might be curious to see those results.

Figure 7: Maybe I am mistaken and if so I apologize in advance but when I compare figure  5 and figure 7 I see some differences that I cannot explain.  a) I think Patient 17 is missing from figure 7. 

b) some patients' (e.g. patient 9) rel. abundance of staph. do not match in figure 5 and 7.  Please verify that there is no mistake. 

Lines 299-301: Since those factors can influence the microbiome, as previously describe, these are confounding factors.  Therefore, I wonder why these factors (or at least the indoor/outdoor living environment) were not in the inclusion criteria to avoid confounding your main result, which is the PBM treatment effect. Is it possible that your objective (treated vs control) would have significant differences if you did not have those confounding factors? Especially the indoors-outdoors?  I would suggest you repeat the analysis for the indoor dogs and outdoor dogs separately (treated vs. non-treated area) and see if alpha and beta diversity are significantly different. I think that will allow overcoming any obstacles introduced by the confounding factors. And it would add a lot of value to the discussion. 

Line 302: References listed here are not complete. There some more studies done evaluating those factors. I would suggest you add those too.

Line 349: here as through the study you are referring tho the MLS Laser therapy. If I understood correctly this instrument has different protocols (Hz, wavelengths etc.) But you used a specific protocol for your study. Please change the language by specifying the protocol's characteristics (e.g. wavelength, mode, minutes etc) and then in brackets have the name of the instrument and the manufacturer. 

Lines 350 - 351: I think this is an overstatement. Please correct. The same for the summary and the abstract. 

Lines 351- 354  This is also an overstatement. Please change and delete the word well. 

Round 2

Reviewer 1 Report

Comments and Suggestions for Authors

I would like to thank the Authors for the effort they put on revising their manuscript. I saw a significative improvement in the quality of the draft. 

I would like to add a few more things to address, in order to finalise the article:

Line 166: the sentence has been changed but it is still missing a piece. The sentence is: “the swabs contained the “powerbead pro tubes” were processed…” and instead it should be “the swabs contained in the “powerbead pro tubes” were processed…”.

Line 174: V3-V4 hypervariable regions of the 16 rRNA gene.

Line 179: you should add the references of those articles in the text.

Reviewer 2 Report

Comments and Suggestions for Authors

Great work

Author Response

The authors are most grateful of your previous comments and suggestions that improved the quality of the manuscript.